# Development of a Direct Competitive ELISA Kit for Detecting Deoxynivalenol Contamination in Wheat

**DOI:** 10.3390/molecules25010050

**Published:** 2019-12-22

**Authors:** Li Han, Yue-Tao Li, Jin-Qing Jiang, Ren-Feng Li, Guo-Ying Fan, Jun-Mei Lv, Ye Zhou, Wen-Ju Zhang, Zi-Liang Wang

**Affiliations:** 1College of Animal Science and Technology, Shihezi University, Shihezi 832000, China; HanLi3909@163.com; 2College of Animal Science and Veterinary Medicine, Henan Institute of Science and Technology, Xinxiang 453003, China; liyuetao2019@163.com (Y.-T.L.); jjq5678@126.com (J.-Q.J.); lirenfeng2019@163.com (R.-F.L.); fanguoy@163.com (G.-Y.F.); junmeirosa@163.com (J.-M.L.); YeZhou9502@163.com (Y.Z.)

**Keywords:** deoxynivalenol, dcELISA kit, performance measurement, development

## Abstract

This study was conducted to develop a self-assembled direct competitive enzyme-linked immunosorbent assay (dcELISA) kit for the detection of deoxynivalenol (DON) in food and feed grains. Based on the preparation of anti-DON monoclonal antibodies, we established a standard curve with dcELISA and optimized the detection conditions. The performance of the kit was evaluated by comparison with high-performance liquid chromatography (HPLC). The minimum detection limit of DON with the kit was 0.62 ng/mL, the linear range was from 1.0 to 113.24 ng/mL and the half-maximal inhibition concentration (IC_50_) was 6.61 ng/mL in the working buffer; there was a limit of detection (LOD) of 62 ng/g, and the detection range was from 100 to 11324 ng/g in authentic agricultural samples. We examined four samples of wheat bran, wheat flour, corn flour and corn for DON recovery. The average recovery was in the range of 77.1% to 107.0%, and the relative standard deviation (RSD) ranged from 4.2% to 11.9%. In addition, the kit has the advantages of high specificity, good stability, a long effective life and negligible sample matrix interference. Finally, wheat samples from farms in the six provinces of Henan, Anhui, Hebei, Shandong, Jiangsu and Gansu in China were analyzed by the kit. A total of 30 samples were randomly checked (five samples in each province), and the results were in good agreement with the standardized HPLC method. These tests showed that the dcELISA kit had good performance and met relevant technical requirements, and it had the characteristics of accuracy, reliability, convenience and high-throughput screening for DON detection. Therefore, the developed kit is suitable for rapid screening of DON in marketed products.

## 1. Introduction

Deoxynivalenol (DON), also known as vomitoxin, is a highly toxic secondary metabolite produced by *Fusarium graminearum* and *Fusarium culmorum*; DON belongs to the B-group of trichothecenes and widely exists in various agricultural products, food, and animal feed, especially in wheat, maize, and other cereal crops [1,2,3,4]. DON readily acts as an animal antifeedant and shows immunotoxicity, organ toxicity, inhibition of protein synthesis, and teratogenicity. These symptoms are closely related to immune suppression, Keshan disease, oesophageal cancer and other diseases [5,6,7]. Moreover, DON is heat-stable, and general cooking and processing cannot destroy its toxicity. Young et al. [8] found that grain processed into pet food still contained DON. Therefore, DON pollution poses a great threat to human and livestock health and has attracted the attention of countries around the world [9]. At present, at least 100 countries have mandatory limits on DON levels in food and feed. In view of its serious toxic effects, in the preliminary draft of the DON maximum levels (MLs), the Codex Alimentarius Food (FAO) Committee recommended the following limits: 2 mg/kg in unprocessed cereals, 1 mg/kg in semi-processed products using wheat, corn and barley as raw materials, and 0.5 mg/kg in cereals for infants and young children [10,11]. In China, the ML of DON in maize, wheat, and their products is regulated at 1 mg/kg [12,13]. The molecular formula of DON is C_15_H_20_O_6,_ and its molecular weight is 296.32 [14]. Its structure is shown in Figure 1:

At present, the main physical and chemical methods for detecting DON contamination in food and feed are thin-layer chromatography (TLC), high-performance liquid chromatography (HPLC), gas chromatography (GC), mass spectrometry (MS), gas chromatography-mass spectrometry (GC-MS), high-performance liquid chromatography–mass spectrometry (HPLC–MS), high-performance liquid chromatography–tandem mass spectrometry (HPLC–MS/MS), and others [15,16,17]. These methods have high precision and sensitivity, but the sample pretreatment is rigorous, the instruments are expensive, the detection range is small, and, as the analysts often need special training, the cost is high. These methods are only suitable for large enterprises, scientific research institutes, or testing institutions that require high detection sensitivity, and they are not suitable for the demand of DON pollution detection in the feed industry. Therefore, increasing attention has been paid to the simple, rapid, sensitive, low-cost enzyme-linked immunosorbent assay (ELISA), which is suitable for large-scale sample screening. For example, the traditional immunosorbent assay (ELISA) [18], chemiluminescence enzyme immunoassay (CLEIA) [19], fluorescence polarization immunoassay (FPIA) [20], time-resolved fluorescence immunoassay (TRFIA) [21], colloidal gold immunochromatography (GICA) [22,23], surface plasmon resonance (SPR) immunoassay [24], silver-stained GICA [25], nanobody-based ELISA [26], and immunosensor, among others, can be used to detect DON. Therefore, due to the prevalence of DON contamination and the large number of samples that need to be analyzed, ELISA kits have been considered a suitable detection tool, and their development and application has grown rapidly in recent years because they do not need special instruments and equipment, are suitable for the field and are suitable for high-throughput screening. 

The purpose of this experiment is to assemble and optimize a new DON dcELISA kit. The performance of the ELISA kit was tested, and its accuracy was verified by HPLC, which laid a foundation for the development of ELISA kits with high sensitivity, specificity, and good quantification suitable for screening a large number of DON-contaminated samples.

## 2. Materials and Methods

### 2.1. Reagents and Materials

The standards of DON, 3-Ac-DON, 15-Ac-DON, Nivalenol (NIV), Fusarenon-X, T-2 toxin, Zearalenone (ZEN), and Aflatoxin B1 (AFB1) were purchased from Sigma-Aldrich Co., Ltd. (Augsburg, Germany). Bovine serum albumin (BSA), chicken ovalbumin (OVA), *N*,*N*′-carbonyldiimidazole (CDI), anhydrous tetrahydrofuran (THF), *N*,*N*-dimethylformamide (DMF), horseradish peroxidase (HRP), 1-ethyl-3-(3-dimethylamino)propyl) carbodiimide hydrochloride (EDC), Freund’s complete adjuvant (FCA), and Freund’s incomplete adjuvant (FIA) were provided by Pierce. PEG-1500 (polyethylene glycol) was purchased from Roche. GaMIgG was purchased from Huamei Biotechnology Company (Shanghai, China). In addition, 96-well microtiter plates as well as 24-well and 96-well cell culture plates were purchased from Iwaki Co., Ltd. (Dalian, China); 3,3,5,5-Tetramethylbenzidine (TMB), phenacetin and urea peroxide were purchased from Sigma. Foetal bovine serum (FBS) was purchased from Gibco. Female Balb/c mice (6 to 8 weeks old) were provided by Beijing SPF Biotech Co., Ltd. (Beijing, China) and were raised under strict control in our laboratory animal house.

Phosphate-buffered saline (PBS), carbonate-buffered saline (CBS), washing buffer (PBST, PBS containing 0.05% Tween-20), blocking buffer (SPBST, PBST containing 5% goat serum), color substrate solution (TMB), stopping solution (2 M H_2_SO_4_), Glucose sodium chloride potassium chloride solution (GNK), complete medium, Hypoxantin Aminopterin and Thymidin (HAT) medium, Hypoxantin and Thymidin (HT) medium, were all made in-house in our laboratory.

A Galaxy S-type CO_2_ cell incubator was purchased from Biotech. A Multiskan MK3 microplate reader was purchased from Thermo (Waltham, Ma, USA) and used for 450 nm absorbance measurements. An inverted MIC 00949 microscope was purchased from Nikon Corporation. A DK-8D water bath was provided by Yiheng Instrument Co., Ltd. (Shanghai, China). A BS124S electronic balance was purchased from the German Sartorius Group. Purified water was prepared using a Milli-Q purification system (Millipore Corporation, Bedford, MA, USA). An A11 basic analytical mill was provided by IKA (Staufen, Germany). A Legend Micro 17 centrifuge was provided by Thermo (Waltham, MA, USA). Glass microfiber filter paper was purchased from Whatman (Maidstone, UK). The reliability of the ELISA kit was confirmed using an Agilent 1260 HPLC equipped with a diode array detector (DAD) (Agilent Technologies, Wilmington, DC, USA).

### 2.2. Preparation of the Antigen and Anti-DON Monoclonal Antibody (mAb)

According to the molecular structure of DON, the artificial antigen DON-BSA was synthesized by the carbonyl diimidazole (CDI) procedure outlined in a previously published method by Maragos et al. [14], with slight modifications. The synthesis of coated DON-OVA was improved by referring to the method of Li et al. [27]. DON was derivatized by maleic anhydride, and then the hapten was coupled with OVA to coat the original DON-OVA by implementation of the carbodiimide (EDC) procedure. Preparation of anti-DON mAb was achieved with classical hybridoma technology [28]. After obtaining DON mAb hybridoma cell lines, this experiment adopted an in vivo induced ascites method [29] to mass produce DON mAb, which was then purified from ascites by an octanoic acid/ammonium sulfate precipitation method [30]. The DON mAb was then stored at −20 °C until the dcELISA kit was assembled.

### 2.3. Development of the DON dcELISA Kit

We prepared the enzyme-labeled hapten (horseradish peroxidise-DON) and determined its working concentration as follows.

The enzyme-labeled hapten (HRP-DON) was prepared by a carbonyl diimidazole (CDI) method. DON standard (5 mg) was dissolved in 1 mL THF, 60 mg CDI was added, and the reaction proceeded for 4 h in a dry environment at 70 °C. The solvent of the reaction products was evaporated, and 500 μL DMF was added to the remaining products and completely dissolved. Then, 2 mg of HRP was added dropwise (the HRP was dissolved in 2 mL 0.01 mol/L pH 7.4 PBS solution) and stirred for 24 h at 4 °C in the dark. The reaction products were dialyzed in PBS for 72 h, the fluids were replaced 9 times during dialysis, and the dialysate, which was the enzyme-labeled hapten HRP-DON, was collected. 

It is well known that the working concentration of the coated antigen and antibody is the key to determine the sensitivity of an ELISA kit. To determine the optimum dilution of RaMIgG, anti-DON mAb, and HRP-DON, chessboard titration tests were carried out. HRP-DON was added to 50% glycerol and stored at −20 °C.

### 2.4. Components of the ELISA Kit

The optimum conditions of the kit were very important for improving detection technology. The components and parameters of the kit are shown in Table 1 [31]: 

### 2.5. Establishment of the Kit Standard Curve

The standard curve was established by dcELISA. The inhibition rate B/B_0_ of a series of concentrations of DON standards against DON mAb was taken as the ordinate, and the logarithmic value of a series of concentrations of DON standards was taken as the abscissa. The standard curve was analyzed and fitted using Origin Program 7.0 software (OriginLab Co., Northampton, MA, USA), and the linear regression was established. The theoretical detection limit and linear detection range of the kit were calculated by the regression equation. 

### 2.6. Pretreatment of Samples

The wheat samples came from farms in six Chinese provinces: Henan, Anhui, Hebei, Shandong, Jiangsu and Gansu. A total of 30 samples were randomly checked (5 samples from each province). After the samples were ground, 5 g of each sample was accurately weighed (accurate to 0.01 g) and placed in a bottle. Distilled water (25 mL) was added, and extracted by sonication for 10 min. The mixture was evenly mixed for a few minutes. The supernatant was centrifuged at 8000 rpm/min for 5 min. Finally, 500 μL of the supernatant was added to 500 μL of the sample diluent, which is the extract solution of the sample to be tested. In addition, the pH of the sample extract was adjusted from 6 to 8. If needed, samples were diluted with the working buffer before being analyzed with the kit.

### 2.7. Operating Procedure of the Kit

(1)Addition of anti-DON mAb: anti-DON mAb at a working concentration was added (50 μL/well), set as the negative and blank control, incubated for 15 min at 37 °C, and then washed.(2)Addition of HRP-DON and the sample to be tested: HRP-DON at a working concentration was added (50 μL/well), the sample to be tested was added at the same volume, the plate was incubated for 25 min at 37 °C, and then washed.(3)Coloration: the TMB-containing color substrate solution (50 μL/well) was added, and the plate was placed in the dark for 5 min at room temperature (RT).(4)Termination: a 2 mol/L H_2_SO_4_ termination solution was added (50 μL/well).(5)Measurement with the microplate reader: the absorbance was measured at 450 nm, and the inhibition rate was calculated.

### 2.8. Characteristics of the DON dcELISA Kit

#### 2.8.1. Sensitivity Determination

According to the method of Hayashi et al. [32], the sensitivity of competitive ELISA is B/B_0_% = 83.3%; the sensitivity of the kit was calculated according to the standard curve regression equation, and the detection limit was also determined. 

#### 2.8.2. Accuracy and Precision Determination

In this study, the accuracy and precision of the kit were determined with recovery experiments and expressed as recovery (%) and relative standard deviation (RSD%), respectively. The wheat bran, wheat flour, maize flour, and maize were first treated with 1% Na_2_CO_3_ for detoxification [33]. Then, 5 g of each sample was spiked with DON at 200, 500 and 1000 ng/g and stirred for 2 h at room temperature (RT). Next, the spiked samples were added to 10 mL of working buffer containing 20% methanol, and extracted by sonication for 10 min. The supernatant was centrifuged at 8,000 rpm/min for 5 min. Finally, 500 μL of the supernatant was added to 500 μL of the working buffer, which is the extract solution of the sample to be tested. Then, each sample was tested three times, and the recovery (%) and RSD% were calculated:Recovery (%) = the measured value/the actual added value × 100%(1)
(2)$$\mathrm{RSD}\;(\%)=\mathrm{SD}\;(\mathrm{standard}\;\mathrm{deviation})/\overset{\text{¯}}{X}\;(\mathrm{mean}\;\mathrm{value})\times 100\%$$

#### 2.8.3. Specificity Determination

The specificity of the cross-reactions between the kit and other mycotoxins was evaluated, and the formula of cross-reaction rate (CR%) is [34]:CR (%) = [IC_50_ (DON)/IC_50_ (Structural Analogue)] × 100%(3)

#### 2.8.4. Stability Determination

The stability of the kit was evaluated by the changes in B_0_ (the value of absorbance without the DON standard) and B/B_0_ (%) (the ratio value of absorbance with 5 ng/mL DON and without the DON standard) during storage (2 to 8 °C).

#### 2.8.5. Matrix Effect Determination

To analyze the effect of the sample matrix on the sensitivity of the kit, the DON standard solution was dissolved in four samples of wheat bran, wheat flour, corn flour, and corn. These samples were then diluted with sample diluent, and the curve was generated according to the operation of the kit.

### 2.9. Confirmation of the DON dcELISA Kit with HPLC

The wheat samples from farms in the six provinces of Henan, Anhui, Hebei, Shandong, Jiangsu and Gansu in China were tested using the assembled DON dcELISA kit and HPLC. A total of 30 samples were randomly checked (5 samples from each province), and the correlation between the kit and HPLC was evaluated by comparing the results of detection [35]. Sample extraction and HPLC analysis were performed according to the method of the national standard of China GB5009.111-2016 [36], with slight modifications. After the samples were ground, 5 g of each sample was accurately weighed (accurate to 0.01 g) and placed in a clean and capped wide-mouth bottle. Twenty-five milliliters of acetonitrile-H_2_O (20:80, *v*/*v*) and 2 g polyethylene glycol were added. The bottle was capped and extracted by sonication for 30 min. The mixture was evenly mixed for a few minutes. The samples were centrifuged at 6000 rpm/min for 10 min. Finally, the supernatants were filtered through glass microfiber filters to clarify the extract solution of the sample to be tested. Then, the supernatants were purified through DON immunoaffinity columns. The extracted phases were collected and analyzed by HPLC. The HPLC analysis was performed using an Agilent 1260 HPLC equipped with a diode array detector (DAD). Separation was performed on a C18 liquid chromatographic column (150 mm × 4.6 mm × 5 μm) or equivalent, the mobile phase was methanol:water (20:80, *v*/*v*), the flow rate was 0.8 mL/min, the column temperature was 35 °C, the injection volume was 50 μL, and the detection wavelength was 218 nm.

## 3. Results

### 3.1. Development of the DON dcELISA Kit

For the determination of the working concentrations of RaMIgG, anti-DON mAb and HRP-DON using the chessboard titration tests, ELISA microplates were coated with 10 ng/mL RaMIgG. The working concentrations of the anti-DON mAb and HRP-DON were determined as 1:6400 (1.56 ng/mL) and 1:800 (28.5 ng/mL), respectively, when the value of B_0_ reached 1.0.

The key parameters were studied to guarantee the ideal sensitivity and performance of the kit for detecting DON. Under the criteria of a higher value of B_0_/half-maximal inhibition concentration (IC_50_) and lower value of IC_50_, the working buffer, which could greatly affect the sensitivity of the kit, was adjusted. Finally, 5% methanol, 0.5 mol/L Na^+^, and pH 7.4 in the working buffer were selected as the optimal working buffer for the kit (Table 2).

### 3.2. Generating and Fitting the Standard Curve of the Kit

The standard curve of the kit is shown in Figure 2. By analyzing the curve, the regression equation y = −32.433x + 76.608, correlation coefficient R^2^ = 0.972, and IC_50_ = 6.61 ng/mL was obtained, and the detection range (IC_10_ to IC_80_) was 1.0 to 113.24 ng/g.

### 3.3. Performance Measurements of the Kit

#### 3.3.1. Sensitivity Determination

When B/B_0_ = 83.3%, the corresponding DON concentration was 0.62 ng/g, indicating a sensitivity of 0.62 ng/g, which was obtained by substituting the B/B_0_ value into the standard curve regression equation. However, considering the need for positive detection and the error of user operation, the detection limit of the competitive ELISA kit was determined to be 1.0 ng/g.

#### 3.3.2. Accuracy and Precision Measurement

Table 3 shows the four feed samples of wheat bran, wheat flour, corn flour, and corn with the recoveries. The average recovery was in the range of 77.1% to 107.0%, and the RSD ranged from 4.2% to 11.9%. The dcELISA kit meets the requirements of national accuracy and precision, indicating that the kit can be used for the detection of actual samples.

#### 3.3.3. Specificity Determination

Table 4 shows that the cross-reactions between the kit and other mycotoxins were negligible. The cross-reaction rate with 3-Ac-DON was 4.7% and that with other mycotoxins was less than 0.2%, indicating that the kit has high specificity.

#### 3.3.4. Stability Determination

As shown in Figure 3, the values of B_0_ (the value of absorbance without DON standard) and B/B_0_ (%) (the ratio value of absorbance with 5 ng/mL DON and without DON standard) showed acceptable decreases during storage. The results showed that the kit had good stability and that its effective life was at least 12 months.

#### 3.3.5. Matrix Effect Determination

As shown in Figure 4, the curves of the spiked samples of wheat bran, wheat flour, corn flour and corn were close to the DON standard curve by dilution of the extract solution multiple times, and their IC_50_ values were 8.81, 7.59, 6.22 and 5.7 ng/mL, respectively, indicating that the matrix interference was negligible. Therefore, the kit is functional for different substrates and can be used for detecting subsequent samples.

### 3.4. Confirmation of the DON dcELISA Kit with HPLC

Table 5 shows that a total of 30 wheat samples from different provinces in China were tested using the assembled DON dcELISA kit and HPLC. The average value of detection with HPLC was in the range of 560.4 to 1049.1 ng/g, and the RSD ranged from 12.4% to 43.4% (the results of HPLC were corrected by a recovery of 85.7%). The average value of detection with the kit was in the range of 580.5 to 1020.3 ng/g, and the RSD ranged from 13% to 43.8%. The results showed that the test results of the kit were generally higher than those of HPLC. However, the test results of the kit in its linear range were in good agreement with those of HPLC.

Thirty samples of wheat were detected with the kit, 22 of which were found to contain DON, with a concentration range of 254.7to 1258.4 ng/g. Four of the 30 samples were false suspect, with a false suspect rate of 13.3% (Table 6).

## 4. Discussions

### 4.1. Pretreatment of Biotoxin Samples

The pretreatment of samples increases the accuracy of HPLC and ELISA analyses. The samples were extracted and detected for DON; the pretreatment of samples for detection by ELISA was relatively simple, and direct filtration after extraction was sufficient for detection, while the pretreatment of samples for detection with HPLC required an immunoaffinity column. Yang et al. reported [37] that, the recovery rate of DON in ELISA was higher than 75%, and the RSD was 4.7% to 10.6% after the sample was filtered directly, while after passing through the immunoaffinity column, the recoveries of DON in HPLC and ELISA were the same when the spiked concentration of the standard was higher. It is concluded that the DON kit simplifies the process of sample pretreatment and purification. The results are accurate and reliable, and the detection steps are simple. It is very suitable for the rapid detection of a large number of samples. Moreover, ELISA detection technology has the advantages of limited interference, strong specificity, and short enzymatic reaction times, which shortens the whole detection time.

### 4.2. Determination of dcELISA Kit Performance

In this study, the dcELISA kit was assembled with an in-house-developed homemade high-affinity anti-DON monoclonal antibody. The kit performance metrics included sensitivity, accuracy, precision, specificity, stability and matrix effect, among others. Sensitivity determination can be calculated according to the method of Hayashi et al. [32]. The sensitivity of competitive ELISA is B/B_0_ = 83.3%, which can also be calculated by the formula of limit of detection LOD (%) = [(X − 2SD)/X] × 100%. The method of B/B_0_ = 83.3% was adopted in this experiment. The sensitivity was determined as 0.62 ng/mL, the detection limit was 1.0 ng/mL, and the detection range (IC_10_ to IC_80_) was 1.0 to 113.24 ng/mL in the working buffer. According to the procedures of authentic sample pretreatment and extraction, the DON levels of samples were equivalent to a 100-fold dilution and the matrix effects were negligible. Thus, for the analysis of wheat samples, with a sensitivity of 62 ng/g, an LOD of 100 ng/g, and a detection range from 100 to 11,324 ng/g in authentic agricultural samples, the cross-reaction rate with 3-Ac-DON and 15-Ac-DON was 4.7%, less than 0.2%, respectively. The DON ELISA method established by the Ministry of Health in China has a detection limit of 5 ng/mL, and the detection range was 5 to 1000 ng/mL. It had been approved as the national recommended standard detection method of China. Therefore, the DON dcELISA kit assembled in our laboratory meets the domestic detection range and sensitivity standard requirements of DON analysis in food and feed. Compared with the commercial kits, the sensitivity and specificity is higher (3 ng/mL in the working buffer), the DON levels of samples were equivalent to a 100-fold dilution with an LOD of 300 ng/g in authentic agricultural samples, and the cross-reaction rate with 3-Ac-DON and 15-Ac-DON was less than 70%, less than 1%, respectively. Compared with the kit that was developed by Li et al. [12], it has higher sensitivity and specificity (4.9 ng/mL in the working buffer), the DON levels of samples were equivalent to a 40-fold dilution with an LOD of 200 ng/g in authentic agricultural samples, and the cross-reaction rate with 3-Ac-DON and 15-Ac-DON was 5.7%, less than 0.5%, respectively. Accuracy and precision are measured by spiked sample recovery (%) and relative standard deviation (RSD%). Generally, the recovery rate is between 70% and 140%. The average recovery was in the range of 77.1% to 107.0%, and the RSD was 4.2% to 11.9% in this experiment, which meets the national accuracy and precision test requirements, indicating that the kit could be used for the detection of actual samples. Therefore, by evaluating the recoveries and determining the DON content in wheat samples, it is proved that the developed dcELISA kit is accurate, reliable, and simple, and that it requires less instrumentation, and involves simple experimental steps for detecting DON content in food and feed. Compared with commercial kits, it is a more advanced detection method in China and abroad, providing a highly sensitive, economical and safe DON detection method.

### 4.3. Comparison of the Results of the dcELISA Kit and HPLC

The kit test results were generally higher than those of HPLC. The wheat samples from the farms in the six provinces of Henan, Anhui, Hebei, Shandong, Jiangsu and Gansu in China were analyzed for DON content using both the kit and HPLC. A total of 30 samples were randomly checked (five samples from each province). The average value of detection with HPLC was in the range of 560.4 to 1049.1 ng/g, and the RSD ranged from 12.4% to 43.4%. The average value of detection with the kit was in the range of 580.5 to 1020.3 ng/g, and the RSD ranged from 13% to 43.8%. Therefore, the results showed that the test results of the kit were generally higher than those of the HPLC. However, the test results of the kit in its linear range were in good agreement with those of HPLC. Therefore, the kit can be used for the determination of DON in food and feed. Antibodies are the basis of the ELISA kit detection method, which may lead to false positive or false negative results, while HPLC is commonly used as an accurate verification method. There were two main reasons why the kit test results were generally too high. First, high or low pH of the sample solution will affect the test results. Therefore, it is necessary to adjust the pH of the sample solution before detection. Some studies have found that when the pH of a sample extract is lower than 5, the structure of the enzyme in an enzyme-labeled antigen changes irreversibly, and most of its activity is lost, resulting in a reduced color reaction, which leads to false-positive results. The pH of the extracted solution was from 6 to 8 when the samples were purified in this experiment, which met the detection requirements of the kit. Therefore, the pH of the samples did not need to be adjusted when samples were detected. Second, the loss of sample in the pretreatment process of HPLC leads to low detection results (the results of HPLC in this experiment were corrected by a recovery of 85.7%).

## 5. Conclusions

In this experiment, a dcELISA kit method was established by using an anti-DON monoclonal antibody developed in our laboratory. The working concentrations of RaMIgG, anti-DON mAb and HRP-DON were optimized, and the performance of the developed kit was tested. Finally, a comparison of the results of the kit with those of HPLC shows that the developed kit has the same detection ability as HPLC. Therefore, the kit can be widely used for DON detection in food and feed.

## Figures and Tables

**Figure 1 molecules-25-00050-f001:**
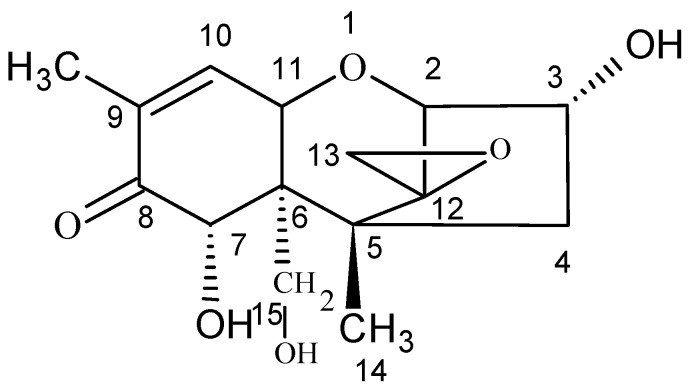
Molecular structure of deoxynivalenol (DON).

**Figure 2 molecules-25-00050-f002:**
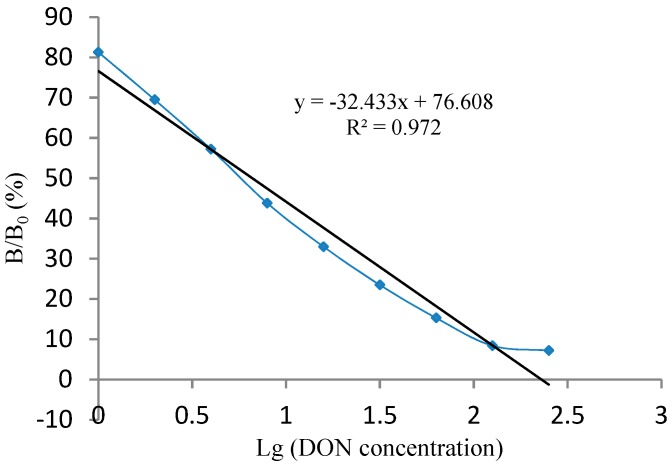
Calibration curve of the dcELISA kit.

**Figure 3 molecules-25-00050-f003:**
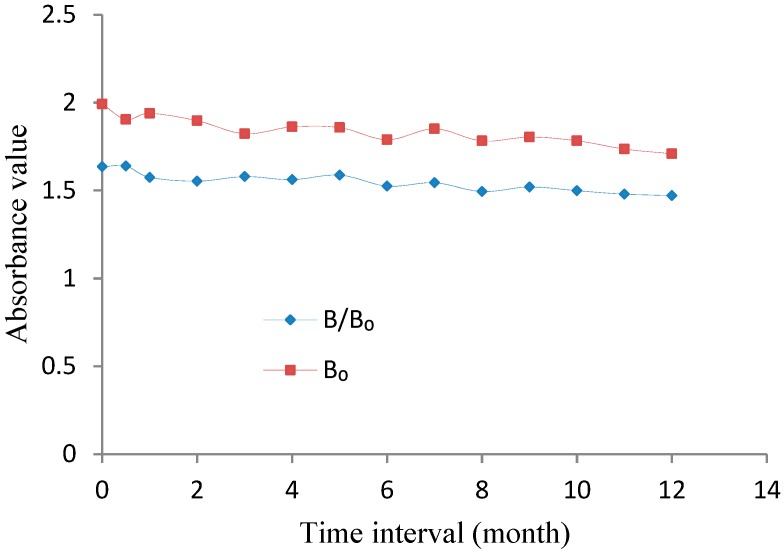
Stability of the dcELISA kit.

**Figure 4 molecules-25-00050-f004:**
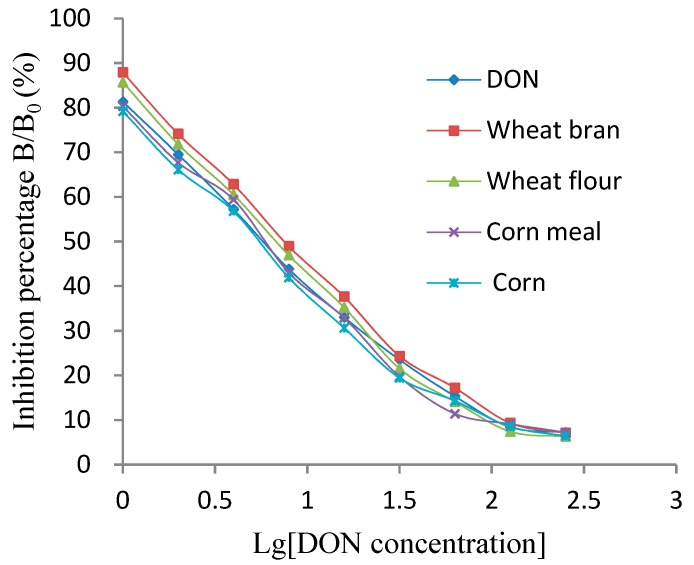
Effect of different samples’ matrixes on the dcELISA kit.

**Table 1 molecules-25-00050-t001:** Components and parameters of the direct competitive enzyme-linked immunosorbent assay (dcELISA) kit.

Number	Composition	Quantity	Unit	Parameters
1	ELISA microplates	1	board	coated 96-well transparent microplates
2	Mab solution	1	tube	6 mL (concentration of 1:6400 diluted in PBS)
3	DON standard	1	tube	1 mg/mL
4	HRP-DON	1	tube	3 mL (concentration of 1:800 diluted in PBS)
5	1 × Working buffer	1	bottle	50 mL (5% methanol, 0.5 mol/L Na^+^, pH 7.4 in PBS)
6	10 × Washing buffer	1	bottle	50 mL (10 × PBST, pH 7.4)
7	Color substrate buffer	1	bottle	15 mL (0.4 mmol/L TMB and 3 mmol/L H_2_O_2_ diluted in citrate buffer, pH 5.0)
8	Stop buffer	1	bottle	10 mL (2 mol/L H_2_SO_4_ diluted in H_2_O)

**Table 2 molecules-25-00050-t002:** Key parameters for the proposed kit.

Factor	Parameter	Factor	Parameter
RaMIgG	10 ng/mL	Methanol (*v*/*v*, %)	5
anti-DON mAb	1:6400 (1.56 ng/mL)	Na^+^ (mol/L)	0.5
HRP-DON	1:800 (28.5 ng/mL)	pH	7.4

**Table 3 molecules-25-00050-t003:** Recoveries of DON in different samples by the dcELISA kit (*n* = 3).

Samples		Spiked (ng/g)	Mean Recovery ± SD (%)	RSD (%)
		200	89.2 ± 6.2	7
Wheat bran		500	88.1 ± 5.7	6.5
		1000	79.4 ± 7.5	9.4
		200	77.1 ± 9.2	11.9
Wheat flour		500	81.7 ± 5.6	6.8
		1000	96.5 ± 4.1	4.2
		200	104.4 ± 5.8	5.5
Corn meal		500	96.4 ± 6.3	6.5
		1000	107.0 ± 7.6	7.1
		200	103.7 ± 4.6	4.4
Corn		500	95.0 ± 5.3	5.6
		1000	98.4 ± 7.3	7.4

**Table 4 molecules-25-00050-t004:** Cross-reactivity of the DON dcELISA kit with other related mycotoxins.

Compounds	IC_50_ (ng/mL)	Cross-Reactivity (%)
DON	6.61	100
3-Ac-DON	142.1	4.7
15-Ac-DON	>5 × 10^3^	<0.2
DON-3-G	>1 × 10^4^	<0.1
NIV	>1 × 10^4^	<0.1
Fusarenon-X	>1 × 10^4^	<0.1
T-2 toxin	>1 × 10^4^	<0.1
ZEN	>1 × 10^4^	<0.1
AFB1	>1 × 10^4^	<0.1

**Table 5 molecules-25-00050-t005:** Comparison of screening results of 30 wheat samples detected by two different methods.

Province	dcELISA Kit			HPLC		
	Range (ng/g)	Average Value ± SD%	RSD (%)	Range (ng/g)	Average Value ± SD%	RSD (%)
Henan	309.5–1243.8	933.4 ± 324.1	34.7	277.1–1231.4	918.1 ± 304.7	33.2
Anhui	378.9–1230.7	810.5 ± 276.3	34.1	350.5–1218.4	796.6 ± 268.5	33.7
Hebei	696.4–1087.9	853 ± 119.4	14	681.9–1023.1	834.3 ± 103.1	12.4
Shandong	741.9–1258.4	1020.3 ± 168.2	16.5	752.8–1235.6	1049.1 ± 158.2	15.1
Jiangsu	591.2–891.2	713.4 ± 93	13	583.5–870.3	703.9 ± 92.4	13.1
Gansu	254.7–1080.5	580.5 ± 254.3	43.8	236.1–991.6	560.4 ± 243.3	43.4

**Table 6 molecules-25-00050-t006:** Test results of wheat samples from different provinces by the kit and HPLC.

Province	Samples (Wheat)	dcELISA Kit	HPLC
	1	+	+
	2	+	+
Henan	3	-	-
	4	+	+
	5	+	**-**
	1	+	+
	2	-	**+**
Anhui	3	+	**-**
	4	+	+
	5	+	+
	1	+	+
	2	+	+
Hebei	3	-	-
	4	+	+
	5	-	**+**
	1	-	-
	2	+	+
Shandong	3	+	+
	4	+	+
	5	-	-
	1	+	+
	2	+	+
Jiangsu	3	-	-
	4	-	-
	5	+	+
	1	+	**-**
	2	+	+
Gansu	3	+	+
	4	+	+
	5	+	**-**

+, positive; -, negative; **+**, false negative; **-**, false suspect.

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
