# Peer review of "Development of a Direct Competitive ELISA Kit for Detecting Deoxynivalenol Contamination in Wheat"

_molecules, 2019, doi:10.3390/molecules25010050_

Round 1
Reviewer 1 Report
Mycotoxin contamination in food and feeds are a serious problem. Sensitive detection is important. This manuscript describes the direct competitive ELISA kit developed by the authors. The results seemed comparable to HPLC method, yet negating the tedious processing and the expensive equipment. Some language minor revisions are required before acceptance.

Author Response
please see the attacment.

Reviewer 2 Report
General Comments:
The manuscript describes a new competitive direct ELISA method for the detection/quantitation of the mycotoxin, deoxynivalenol. It claims to have developed a new, more sensitive antibody with lower LOD than commercial DON ELISA kits. As such, it appears to provide an improvement over existing DON ELISA methods/kits that are presently on the market. However, the authors don’t actually compare the data from commercial kits to their new antibody on the same 30 or so wheat samples (see section 4.2). They do, however, compare the results to conventional HPLC analysis (see section 4.3) which is a legitimate comparison and they account for the general trend that the antibody results are generally a bit higher than those of HPLC.
The manuscript is generally well written with few errors or typos. The description of the methodology is extensive and technical and appears to be sound. There are not an extensive number of Figures (4) and Tables (3) but these appear to cover all aspects of the validation (matrix effects, calibration, cross-reactivity, stability etc.). Whether the method was developed solely for internal use in China or to be commercialized is not clear. The introduction covers the various different immune based assays for DON, but does not really describe the various commercial kits or the need to develop yet another DON antibody, other than perhaps increasing the sensitivity or reducing the costs per sample of commercial kits. The quest for sensitivity seems to be at the expense of cross-reactivity – this antibody is highly specific for DON but fails to detect other DON adducts – 3-ADON, 15-ADON and DON-3G – which are gaining importance in Europe as “total DON” measurement. While the method appears to be more sensitive than others for DON, how great is this requirement, considering the values of established safe limits of DON in various cereals?
Specific comments:
Page 2, line 60 – Are these the correct references (15-17) for this particular sentence?
Page 4, Table 1. Please adjust the width of the Table such that the headings and entries are continuous and not split into two lines.
Page 5, Line 162 – “plate”?
Page 7, Table 3. It would be interesting to explore why this antibody is so highly specific to DON as compared to other commercial antibody kits. However, this may not be such a desirable trait as efforts should be made to find an antibody that can measure “total DON”, including adducts such as DON-3G (see above).
Page 9, section 4.2. While it might be suggested that the authors should have done a direct comparison of their antibody method to 2 or 3 commercial kits, here at least they could expand on the last sentence to show how their system compares. Most commercial kits quote a limit of detection and cross reactivity.
Page 9, line 302-308. I am not sure I understand the pH argument. I don’t see why an extract using 25 ml of 80% H2O/20% MeOH of a 5 gm grain sample would be acidic to the point of a pH of 5. What was the average pH before adjusting to 6-8? I do, however, accept, the loss of sample during cleanup before HPLC.
Reviewer 3 Report
Abstract:
Lines 14-15: The detection limit and lineal range of DON by the kit should be expressed in food basis (i.e. µg DON/kg cereal) (again in lines 215-218 and 274-275)
Introduction
Line 35-36: What do you mean by the sentence ‘DON belongs to the B-group of monofilamentous compounds’? DON belongs to the B-group of trichothecenes.
Line 37: ‘their production by-products’ is redundant, please revise the standard of English throughout manuscript.
Line 40: DON is not usually related to oesophageal cancer, while fumonisins are.
Lines 46 and 50: Please express maximum levels in µg/kg or mg/kg.
Line 75: Please put the objectives of the work in a separate paragraph.
Material and methods
Line 82: Zearalenone is usually abbreviated as ZEN or ZEA (not ZER)
Lines 105-107: HPLC conditions and analytical method are missing
Line 122/176: Why DON standard was dissolved in anhydrous tetrahydrofuran (THF) for spiking purposes? Other solvents are much more common.
2.6. Pretreatment of samples. It is taking too much time (3 hours approx.) so the method is nor as rapid and simple as compared to instrumental methods (LC-based)(see lines 255-258)
Line 154: What is ‘sample diluent’? Please specify.
2.8.2. Accuracy and precision determination. It is unclear the treatment used for spiked samples. What was spiked with DON, the solid cereal sample before extraction (correct) or the extracted liquid (not good)? It is deduced that samples were spiked and maintained for 2 hours at RT (define all abbreviations), and after they were subjected to pretreatment (as described in 2.6) for another 3 hours? , please specify. Also, provide a reference for the use of Na2CO3 for detoxification.
2.9. Confirmation of the DON dcELISA kit with HPLC. Please briefly describe the LC-based analytical method (extraction and purification steps). Also, indicate performance parameters of the LC-method: recovery and limit of detection (LOD).
Results
Table 2. Why express the DON spiking level in cereal samples as ng/mL? The samples should have been spiked before extraction. This raises the question of what range of concentrations (in food basis) are actually been measured by the kit.
Line 248: expression units for DON concentrations are missing
Table 4. It should be stated if DON levels were corrected for recoveries.
According to the standard methodology for validation of ELISA method, there are several important parameters that are missing in the manuscript:
determination of the cut-off value and determination of the false negative and false suspect rate. Negative control (blank matrix) sample: a sample known to be free of the mycotoxin to be screened for, e.g. by previous determination using a confirmatory method of sufficient sensitivity.Discussion
Line 262: The recoveries values of 49.8% and 49.7% do not match with those in table 2, why?
Anyway, valid recoveries for DON by any method should be in the range of 60-120%.
Round 2
Reviewer 3 Report
Lines 306-307: The sentence 'the recoveries were 49.8% and 49.7%, respectively' is misleading because it suggests that the acceptable minimum recovery set at 60% was not reached. Please delete or change.
Comment: The term 'false suspect rate' is the same as 'false positive rate'
Usually, in screening methods (such as dcELISA), the result of the screening analysis is either ‘negative’ or ‘suspect’. ‘Suspect sample’ (screen positive) means the sample exceeds the cut- off level and may contain the mycotoxin at a level higher than the screening target concentration. Any suspect result triggers a confirmatory analysis for unambiguous identification and quantification of the mycotoxin. Then, ‘False suspect sample’ is a negative sample that has been identified as suspect.
